# MEMORIZE TO FORGET: MACHINE UNLEARNING WITH-OUT GRADIENT ASCENT VIA MODEL EXTRAPOLATION

## ABSTRACT

For ethical and safe AI, machine unlearning rises as a critical topic aiming to protect sensitive, private, and copyrighted knowledge from misuse. To achieve this goal, it is common to conduct gradient ascent (GA) to reverse the training on undesired data. However, such a reversal is prone to catastrophic collapse, which leads to serious performance degradation in general tasks. As a solution, we propose *model extrapolation* as an alternative to GA, which reaches the counterpart direction in the hypothesis space from one model given another reference model. Therefore, we leverage the original model as the reference, further train it to memorize undesired data while keeping prediction consistency on the rest retained data, to obtain a *memorization model*. Counterfactual as it might sound, a *forget model* can be obtained via extrapolation from the memorization model to the reference model. Hence, we avoid directly acquiring the forget model using GA, but proceed with gradient descent for the memorization model, which successfully stabilizes the machine unlearning process. Our model extrapolation is simple and efficient to implement, and it can also effectively converge throughout training to achieve improved unlearning performance.

## 1 INTRODUCTION

The ground-breaking achievement of Large Language Models (LLMs) has significantly boosted both industrial development and human lives. Through the integration of a tremendous amount of data, LLMs perform as knowledge experts to provide general assistance that is specialized to personal requirements. However, such a close machine-human interaction that is based on large-scale data utilization raises serious concerns about privacy, safety, and ethical issues (Li et al., 2024; Motoki et al., 2024; Karamolegkou et al., 2023). Therefore, Machine Unlearning (MU) (Bourtoule et al., 2021; Cao & Yang, 2015; Fan et al., 2023; Jia et al., 2023) has raised abundant attention, aiming to protect private information, remove malicious data, and avoid copyrighted or proprietary knowledge.

One naive way to unlearn a foundation model is to retrain it from scratch without using the undesired data. However, it is unrealistic due to the unaffordable cost of retraining, which might require at least millions of dollars for training LLMs like Llama (Touvron et al., 2023), Phi (Abdin et al., 2024), and OPT (Zhang et al., 2022). For a more adaptive and desirable solution, reversing the training process of LLMs by gradient ascent (GA) (Li et al., 2024; Hsieh et al., 2019; Jia et al., 2023; Chen & Yang, 2023; Wang et al., 2023; Yao et al., 2024b) has drawn a considerable amount of research and industrial attention. Practically, the training dataset is divided into two subsets, namely *retain set* and *forget set*. The retain set is used to maintain the knowledge of desirable data, and the forget set contains the undesired knowledge that needs to be removed. Intuitively, GA achieves MU by maximizing the loss on the forget set, *i.e.*, ascending the gradient on undesired data as a contrast to gradient descent (GD) on desired data.

However, gradient ascent could be problematic in practice. Maini et al. (Maini et al., 2024) demonstrate that even though GA can improve forget quality, it sacrifices the model utility on real-world tasks. Moreover, Zhang et al. (Zhang et al., 2024) show that GA leads to catastrophic collapse, which seriously sabotages the training stability and deviates the models from their initial reference models. As a result, the generalization performance under real-world tasks degrades drastically after a certain stage of training without gaining satisfactory improvement in forget quality. In order to solve the deficiencies of GA, most of the existing works (Rafailov et al., 2023; Maini et al., 2024;

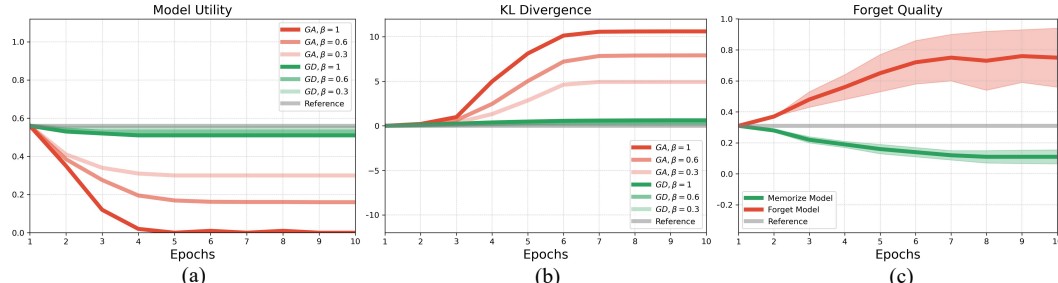

Figure 1: (a) Effect of gradient ascent and gradient descent on model utility under various reweighting levels. (b) Effect of gradient ascent and gradient descent on divergence between training and reference models under various reweighting levels. (c) Comparison of forget quality between the forget model and the memorize model.

Zhang et al., 2024) conduct reweighting for the losses for forgetting and retaining, aiming to trade off the forget quality and modal utility. However, no matter how to reweight the losses, the nature of GA will always pose a negative effect on training stability and model utility.

To further understand the effect of GA, standard GD, as well as the loss reweighting, we conduct an intuitive ablation study on the TOFU benchmark (Maini et al., 2024). Specifically, based on the forget set and the retain set, we can decompose the losses for both GA and GD scenarios. For GA, the learning objective can be formulated as $\mathcal{L}_{GA} = \mathcal{L}_R - \beta\mathcal{L}_F$, where the first loss is for risk minimization on the retain set, and the second loss aims to maximize the standard risk on the forget set. Then for GD, we have $\mathcal{L}_{GD} = \mathcal{L}_R + \beta\mathcal{L}_F$. Note that under both scenarios, we have a weight coefficient $\beta$ to control the importance of the loss on the forget set. We show the model utility on the real-world set from TOFU (Maini et al., 2024) in Figure 1 (a), meanwhile, quantify how training models deviate from the original reference model in Figure 1 (b). We can see that, compared to GD, GA largely degrades and deviates the model from the reference one at any level of weight $\beta$. However, GD is stable throughout the training period. Hence, we ask: Since GA has so many deficiencies compared to standard GD, is it possible to conduct GD without using GA to achieve MU?

In this paper, we propose a counter-strategy that enhances memorization on the forget set using GD to avoid using GA. As a result, the obtained memorization model is the flip side of the forget model that we desire. By leveraging the initial model as a reference, we conduct *MOdel eXtrapolation* (MOX) that reaches the counterpart directions from the memorization model to the reference model. The extrapolation result produces a forget model that lies in the same hypothesis space and can successfully forget the undesired knowledge. As shown in Figure 1 (c), when we encourage memorization, the memorization model shows degraded forget quality, but the counterpart forget model achieves improved forget performance. In this way, we can effectively avoid using GA and stabilize the MU training just using GD. Moreover, the model extrapolation is computationally efficient and can be dynamically deployed throughout the entire training process. Through extensive experiments on well-known benchmarks such as TOFU (Maini et al., 2024) and MUSE (Shi et al., 2024), we carefully validate the effectiveness of our method MOX, by comparing it to state-of-the-art MU approaches. Moreover, we conduct analytical studies to further understand MOX by providing practical insights.

In Summary, our contribution is three-fold:

- We stabilize the MU training by avoiding GA, which provides a novel direction for real-world practices to harmonize forget quality and model utility.

- We propose a novel methodology named model extrapolation, which can be effectively and efficiently deployed into the training process to achieve MU.

- Experimental improvements and practical insights are contributed to advance the development of the MU community.

## 2 RELATED WORKS

MU was taken seriously under several regulatory catalysts, such as the General Data Protection Regulation (GDPR) (Regulation, 2018) and the California Consumer Privacy Act (CCPA) (Bonta, 2022), and was first introduced by Cao & Yang (Cao & Yang, 2015). Pioneering works mainly focused on small-scale models under traditional tasks (Bourtoule et al., 2021; Golatkar et al., 2020; Ginart et al., 2019; Hsieh et al., 2019; Thudi et al., 2022; Izzo et al., 2021; Koh & Liang, 2017; Guo et al.,

2019; Sekhari et al., 2021). Particularly, certified unlearning (Ginart et al., 2019; Bourtoule et al., 2021) provides provably unlearning strategies under certain scenarios, and influence function studies the influence of removing data from training. However, these studies normally require second-order Hessian matrix computation, which is intractable to modern architectures such as LLMs.

Along with the development of foundation models, most research focusing on unlearning has shifted to LLMs (Jang et al., 2022; Wang et al., 2023; Chen & Yang, 2023; Yao et al., 2024b; Li et al., 2024). Specifically, GA (Maini et al., 2024; Hsieh et al., 2019; Zhang et al., 2024; Wang et al., 2025; 2024) is an intuitive way to remove the knowledge that has been learned via GD. The studies proposed variants of GA methods to achieve balanced performance between model utility and forget quality. However, most of the studies can be generalized into loss reweighting, which still keeps the deficiencies of GA, such as training instability and catastrophic collapse. Particularly, Jang et al. (Jang et al., 2022) proposed to maximize the loss of next-token prediction, which successfully achieves MU for LLMs. Yao et al. (Yao et al., 2024a) proposed to combine GA with GD on in-distribution data, which alleviates the negative effect of GA to some extent. Further, NPO (Zhang et al., 2024) proposes to optimize the unlearning model as a weighted preference optimization problem, which only focuses on using the forget set as a negative preference.

Another branch for unlearning is model editing (Guo et al., 2024; Ilharco et al., 2022; Jung et al., 2025; Wu et al., 2023), which aims to operate on the model weight to forget the specific knowledge held in it. Wu et al. (Wu et al., 2023) proposed a Privacy Neuron Detector, which can effectively detect privacy-related neurons to further eliminate their knowledge via a Privacy Editor. However, it is hard to apply the neuron detector to large-scale architectures such as LLMs, and the privacy information is hard to define when the forget set and the retain set are similar to each other. Ilharco et al. (Ilharco et al., 2022) studied task arithmetic, which demonstrates the relationship between tasks from the perspective of hypothesis vectors. If optimization on one task moves the model towards a certain vector, then the negation of the vector in the hypothesis space would lead to forgetting the task. The study was based on relatively small-scale datasets under classification tasks, and it only considered one task during each experimental trial. However, in MU, there are both forgetting and retaining tasks to be fulfilled. Guo et al. (Guo et al., 2024) studied mechanism unlearning, which first localizes tokens that are responsible for extracting facts or knowledge, then it modifies the fact prediction via an MLP layer. However, identifying the tokens and deciding the proper facts is difficult in practice, thus limiting its extension to large-scale MU.

Our MOX approach avoids the deficiencies of GA and stabilizes the unlearning process by only conducting GD, thus showing advantages over GA-based methods. Moreover, MOX resembles task arithmetic, but it is more compatible with large-scale unlearning practices by taking both forgetting and retaining into account. It is also free from identification of certain factors or privacy-related neurons, thus, it is easily extendable to large-scale MU applications.

## 3 METHODOLOGY

In this section, we elucidate the proposed MOdel eXtrapolation (MOX) for MU. First, we formulate the problem setting and with existing baselines. Then, we introduce our methodology, MOX. Finally, we describe the practice of MOX in training and discuss its advantages for unlearning.

### 3.1 PRELIMINARIES

In MU, we are given a retain set $\mathcal{D}_R = \{(x_i, y_i)\}_{i=1}^{m}$ with $m$ instances with each including data $x_i$ and label $y_i$, and a forget set $\mathcal{D}_F = \{(x_i, y_i)\}_{i=m+1}^{m+n}$ which contains $n$ instances. Both $\mathcal{D}_F$ and $\mathcal{D}_R$ belong to our training set $\mathcal{D}$. For MU tasks with no target during the unlearning process, termed non-targeted MU, they only focus on forgetting the knowledge. On the contrary, targeted MU that aims to forget the knowledge, and meanwhile enforces outputting a certain target $\tilde{y}_i$. We denote the unlearning model $h_\theta(\cdot)$, which is parameterized by $\theta$. Under the framework of LLMs, the input data is a sequence of tokens $x = (t_1, t_2, \ldots, t_l)$, which is forwarded into the unlearning model for next-token prediction on $y$: $h_\theta(y|x) = \prod_{i=1}^{l} h_\theta(t_i|t_{<i})$. The learning objective for training LLMs commonly employs Cross-Entropy Loss $\mathcal{L}_{CE}(x, y, \theta) = \log(h_\theta(y|x))$, through which we can obtain the initial model, *i.e.*, reference model $\theta_{ref}$. The performance of MU is commonly evaluated via the prediction quality on both the forget quality and model utility (Maini et al., 2024) from the forget set and the retain set, respectively.

**Gradient Ascent (GA) (Maini et al., 2024)** is a straightforward solution to reverse the training process that minimizes the Cross-Entropy Loss, which is formulated as:

$$\mathcal{L}_{GA} = -\frac{1}{n} \sum_{(x_i,y_i) \in \mathcal{D}_F} \mathcal{L}_{CE}(x_i, y_i, \theta). \tag{1}$$

Intuitively, gradient ascent aims to deviate the learning process from the original direction. Note that the loss is computed only on the forget set; thus, the performance on the retain set would be affected and lead to degradation. Moreover, such a loss maximization is unbounded, which could lead to serious collapse (Maini et al., 2024; Zhang et al., 2024).

**Gradient Ascent with Difference (GAD) (Liu et al., 2022)** improves gradient ascent via introducing a regularization term on the retain set:

$$\mathcal{L}_{GAD} = \frac{1}{m} \sum_{(x_i,y_i) \in \mathcal{D}_R} \mathcal{L}_{CE}(x_i, y_i, \theta) - \frac{1}{n} \sum_{(x_j,y_j) \in \mathcal{D}_F} \mathcal{L}_{CE}(x_j, y_j, \theta). \tag{2}$$

The difference is denoted by the opposite gradient direction between the retain and forget sets. By adding the regularization, it acts as a compromise between forgetting and model utility on the retain set. However, it is hard to balance the two losses in practice. Moreover, the two losses could conflict with each other during training (Maini et al., 2024; Wang et al., 2025).

**Negative Preference Optimization (NPO) (Zhang et al., 2024)** takes inspiration from Direct Preference Optimization (Rafailov et al., 2023), which basically conducts weight GA via an implicit optimization instead of using Cross-Entropy Loss:

$$\mathcal{L}_{NPO} = -\frac{2}{n\beta} \sum_{(x_i,y_i) \in \mathcal{D}_F} \log \sigma(-\beta \log(\frac{h_\theta(y_i|x_i)}{h_{\theta_{ref}}(y_i|x_i)})), \tag{3}$$

where $\beta$ is a temperature hyperparameter, $\sigma(\cdot)$ is the sigmoid function, and $h_{\theta_{ref}}(\cdot)$ denotes the forward function of the reference model. We can see that the temperature acts as a scaling parameter to reweight the importance of each instance. However, it still suffers from inferior model utility and instability due to the effect of GA-based optimization (Wang et al., 2024; 2025).

However, despite the deficiencies of GA, the advantages of the existing method can still be leveraged. Therefore, we propose a counter-strategy that achieves unlearning via memorization.

### 3.2 MEMORIZE TO FORGET

Given the previous findings that using GA for forgetting is problematic, and using GD can only achieve memorization instead of our unlearning goal. Therefore, understanding the relationship between GA and GD is vital to accomplish further "memorize to forget". Ilharco et al. (2022) proposed a *task vector* that demonstrates negating a task vector results in reduced performance on the task, i.e., forgetting the task. Specifically, given a fine-tuned model $f_{new}$, applying a negative vector $-f_{new}$ to the pre-trained model $f$ can extrapolate between $f$ and $f_{new}$. Hence, it is possible to first enhance memorization, then use extrapolation to achieve forgetting, as explained by our MOX below.

### 3.3 MODEL EXTRAPOLATION (MOX)

Based on the above discussion, we suggest a *irreversible gradient* criterion:

**Definition 1** (irreversible gradient). *Given a pre-trained model $\theta$ that has been trained through an optimization task $\Psi$, any downstream fine-tuning task with a gradient that reverses the gradient of $\Psi$ should be avoided.*

The above definition denotes that reversing the gradient via GA is not preferred, which is an important criterion for designing our methodology. The intuition of our methodology is shown in Figure 2. Figure 2 (a) denotes that directly obtaining the forget model $\theta_{for}$ via GA is not feasible in our study due to the irreversible gradient criteria, as $\mathcal{D}_F \subset \mathcal{D}$, and forgetting $\mathcal{D}_F$ goes against the pre-trianing task that aims to fit $\mathcal{D}_F$.

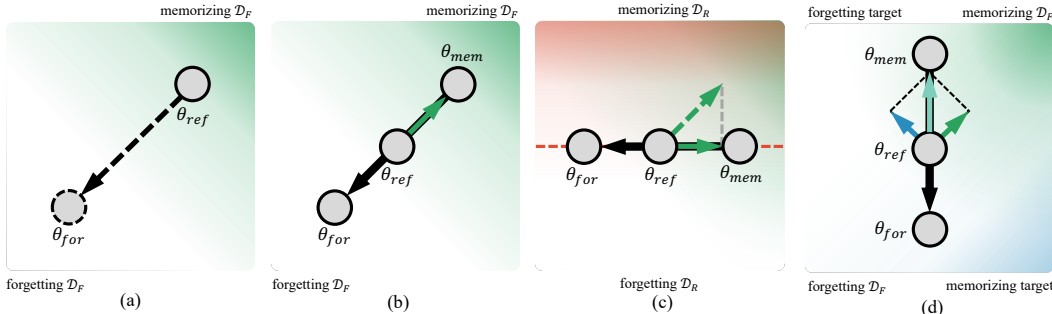

Figure 2: Illustration of our methodology. The color shade level denotes how strongly a model fits a dataset, the colored arrow shows the learning direction, and the black arrow denotes the model extrapolation process. (a) Directly obtaining the forget model $\theta_{for}$ from the reference model $\theta_{ref}$ via GA is infeasible, as it reverses the pre-training process, thus leading to training problems (Section 1). (b) Instead of using GA, we use GD by memorizing $\mathcal{D}_F$ to obtain a memorization model $\theta_{mem}$. Through our model extrapolation, we can produce $\theta_{for}$, which successfully forgets the knowledge from $\mathcal{D}_F$. (c) To further maintain the model utility, we apply the KL-divergence constraint to ensure consistency in model utility performance between $\theta_{ref}$ and $\theta_{mem}$, further benefiting the utility performance of $\theta_{for}$. (d) For targeted MU, we can introduce an additional loss to forget the target knowledge since it does not reverse the pre-training process. By combining it with the memorization loss, we can conduct MOX to obtain $\theta_{for}$ (The KL-divergence constraint is omitted here for simplicity).

Therefore, we propose to intensify memorization on the forget set, and meanwhile maintain the prediction consistency on the retain set between the training model and the reference model. Hence, we can obtain a memorization model $\theta_{mem}$ by leveraging Cross-Entropy Loss:

$$\mathcal{L}_{mem} = \frac{1}{n} \sum_{(x_j, y_j) \in \mathcal{D}_F} \mathcal{L}_{CE}(x_j, y_j, \theta) + \frac{1}{m} \sum_{(x_i, y_i) \in \mathcal{D}_R} \mathrm{KL}(h_{\theta_{ref}}(y|x) \| h_{\theta}(y|x)). \tag{4}$$

Similarly, we can follow NPO to leverage the preference optimization objective:

$$\mathcal{L}_{mem} = \frac{2}{n\beta} \sum_{(x_i, y_i) \in \mathcal{D}_F} \log \sigma(-\beta \log(\frac{h_\theta(y|x)}{h_{\theta_{ref}}(y|x)})) + \frac{1}{m} \sum_{(x_i, y_i) \in \mathcal{D}_R} \mathrm{KL}(h_{\theta_{ref}}(y|x) \| h_\theta(y|x)). \tag{5}$$

In both objectives, the first terms aim to enhance memorization of the forget set. As shown in Figure 2 (b), although we can already obtain a memorization model and effectively obtain a forget model that achieves the forgetting goal (see Section 4), we hope to further maintain the model utility. Hence, we introduce a second term in both Eqs. equation 4 and equation 5 which is the KL-divergence regularization that aims to minimize the Kullback-Leibler (KL) divergence between the predictions on $\mathcal{D}_R$ of the reference model $\theta_{ref}$ and the training model $\theta$. As shown in Figure 2 (c), the learning direction of memorization is constrained to maintain the performance on $\mathcal{D}_R$[1], which helps maintain the general utility performance for the memorization model.

As a result, the obtained model $\theta_{mem}$ retains the preferred knowledge but even fits intensely to the forget set $\mathcal{D}_F$. However, our desired model $\theta_{for}$ should maintain the knowledge from $\mathcal{D}_R$ and forget the undesired knowledge in $\mathcal{D}_F$. Hence, we can reasonably claim that $\theta_{mem}$ is a counterpart of $\theta_{for}$, where the reference model $\theta_{ref}$ plays a critical role between memorizing and forgetting. To further obtain the forget model $\theta_{for}$, we conduct MOX to extrapolate from $\theta_{mem}$ to $\theta_{ref}$ via the following non-convex combination:

$$\theta_{for} := (1 + \alpha)\theta_{ref} - \alpha\theta_{mem}, \ \alpha \in \mathbb{R}^+, \tag{6}$$

where the hyper-parameter $\alpha$ controls the extrapolation strengths for MOX. If $\alpha$ is set to a high value, the forget quality can be enhanced. We show in experiments (Section 4) that the performance improvement of forgetting is significant by reasonably increasing $\alpha$. Although we found that extreme values of $\alpha$ would lead to a decrease in model utility, the decrease is much slower than catastrophic collapse in GA, thus, it is easy to detect and avoid.

Intuitively, by deriving Eq. 6, we have $\theta_{for} := \theta_{ref} + \alpha(\theta_{ref} - \theta_{mem})$, where the task vector $\theta_{ref} - \theta_{mem}$ is scaled by $\alpha$ and added to $\theta_{ref}$. Since the processing of obtaining $\theta_{mem}$ from $\theta_{ref}$

---

[1]Several works (Wang et al., 2025; Maini et al., 2024) aim to achieve enhanced model utility performance on $\mathcal{D}^R$ and forgetting simultaneously. However, we only focus on improving forgetting quality and maintaining model utility, because the utility performance can be effectively enhanced via fine-tuning the pre-trained model (Hu et al., 2023; Zhao et al., 2024). Further, our method is built on the enhanced LLMs for MU without sacrificing the utility performance. Thus, only focusing on forgetting and maintaining utility is sufficient for MU.

is memorizing, the opposite direction $\theta_{ref} - \theta_{mem}$ is forgetting, as demonstrated in Ilharco et al. (2022). Therefore, the difference between our forget model and the reference model $\theta_{for} - \theta_{ref}$ can be interpreted by forgetting the knowledge that $\theta_{mem}$ has memorized.

Our MOX holds several non-negligible advantages compared to some typical MU strategies: 1) Computation Stability: We identify the detrimental failure of GA, and we only use GD for training. Thus, there is no concern for catastrophic collapse or task conflicts. 2) Adaptability: Our learning target can be implemented using various objectives, benefiting from the recent advancement in MU and LLM training. 3) Efficiency: Our MOX is simple to implement, and the forget model can be computed directly from subtracting model parameters, which enables its dynamic deployment during training. Compared to task vectors Ilharco et al. (2022), MOX is more flexible on the extrapolation strength, thus significantly enhancing the forget quality. Additionally, task vectors are limited to only forgetting, which ignores the knowledge retaining goal of machine unlearning. Our approach considers a retain regularization, which largely enhances the model utility. Further, we leverage the advantages and discuss two practical implementations of MOX with advanced effectiveness.

**Targeted Unlearning** is one MU learning problem where a target is provided for examples to be forgotten. As shown in Figure 2 (d), we hope to forget $\mathcal{D}_F$ and memorize the target simultaneously. Although the previous discussion is under non-targeted scenarios, we can still tackle targeted MU by effectively introducing a targeted unlearning term:

$$\mathcal{L} = \mathcal{L}_{mem} - \frac{1}{m} \sum_{(x_i, y_i) \in \mathcal{D}_R} \mathcal{L}_{CE}(x_i, \tilde{y}_i, \theta), \tag{7}$$

where $\tilde{y}_i$ is the target. Similar to Eq. equation 5, we can apply preference optimization for realization. Through training to obtain $\theta_{mem}$, MOX can be applied to targeted MU cases. Note that here directly maximizing the risk does not conflict with the pre-training process, thus it can be effectively applied.

**Momentum Extrapolation** is an effective technique for improving generalization and forget quality, thanks to the computation efficiency of our MOX. Since the extrapolation process can be conducted on-the-fly, we can gradually update the forget model by ensembling historical versions throughout the training:

$$\theta_{for}^t := \eta \theta_{for}^t + (1 - \eta) \theta_{for}^{t-1}, \tag{8}$$

where $\eta$ is a momentum coefficient (He et al., 2020; Tarvainen & Valpola, 2017) and it is normally set to 0.675, and $\theta_{for}^{t-1}$ is the history ensemble. Once obtaining the current $\theta_{for}^t$, we update it via Eq. equation 8 to incorporate historical weight, which largely benefits generalization performance for both forget quality and model utility. Next, we conduct experiments to validate our MOX.

## 4 EXPERIMENTS

In the experiments, we first compare our method with several typical baseline methods to validate its effectiveness. Then, we conduct an ablation study to decompose and investigate each module of our MOX. Moreover, we analyze the hyper-parameters to understand our experimental design. Further, we study the stability of MOX over various datasets with different forget sizes during training.

**Baseline Methods.** We compare MOX to various strong LLM unlearning techniques, namely, Gradient Ascent (GA) (Maini et al., 2024), KL-divergence minimization (KL) (Maini et al., 2024), Gradient Ascent with Difference (GAD) (Liu et al., 2022), NPO (Zhang et al., 2024), AltPO (Mekala et al., 2024), SimNPO (Fan et al., 2024), and RMU (Li et al., 2024), Task Vectors (TV) (Ilharco et al., 2022), LLM Unlearning (LLMU) (Yao et al., 2024b), and Who's Harry Potter (WHP) (Eldan & Russinovich, 2023). Additionally, we add Preference Optimization (PO) (Maini et al., 2024) and Direct Preference Optimization (DPO) (Rafailov et al., 2023) using target data to conduct targeted unlearning. Specifically, we use the template "I don't know the answer" as the target data, as done in Rafailov et al. (2023). For our method, we tune the values of extrapolation strength $\alpha$ as 0.5, 1.0, 2.0, 4.0, and 8.0. Further, we also conduct targeted unlearning for MOX. Additionally, we investigate the performance of MOX with momentum extrapolation to further justify its effectiveness. Our evaluation follows Wang et al. (2024), where the evaluation is reported based on the last epoch, instead of the best performance during training (Zhang et al., 2024).

Table 1: Comparison of MOX and baseline methods on TOFU benchmark. The top two best performances in each column are highlighted in **bold**.

| Base LLM | Llama2-7B | | | | Phi-1.5B | | | |
|---|---|---|---|---|---|---|---|---|
| Metric | FQ($\uparrow$) | MU($\uparrow$) | F-RL($\downarrow$) | R-RL($\uparrow$) | FQ($\uparrow$) | MU($\uparrow$) | F-RL($\downarrow$) | R-RL($\uparrow$) |
| Original LLM | 0.0000 | 0.6346 | 0.9851 | 0.9833 | 0.0013 | 0.5184 | 0.9607 | 0.9199 |
| Retrained LLM | 1.0000 | 0.6267 | 0.4080 | 0.9833 | 1.000 | 0.5233 | 0.4272 | 0.9269 |
| GA | 0.0143 | 0.6333 | 0.4862 | 0.9008 | 0.0213 | 0.5069 | 0.5114 | 0.8048 |
| KL | 0.0168 | 0.6300 | 0.5281 | 0.9398 | 0.0120 | 0.5047 | 0.5059 | 0.8109 |
| GAD | 0.0268 | 0.6320 | 0.4773 | 0.8912 | 0.0215 | 0.5110 | 0.4996 | 0.8496 |
| PO | 0.0541 | 0.6308 | **0.3640** | 0.8811 | 0.0286 | 0.5127 | 0.3170 | 0.7468 |
| LLMU | 0.0541 | 0.6337 | 0.4480 | 0.8865 | 0.0286 | 0.5110 | **0.3058** | 0.7270 |
| DPO | 0.0541 | 0.6359 | 0.5860 | 0.8852 | 0.0521 | 0.5125 | 0.3437 | 0.7349 |
| NPO | 0.0068 | 0.6321 | 0.4632 | 0.8950 | 0.0030 | 0.5057 | 0.5196 | 0.8000 |
| AltPO | 0.0120 | 0.6432 | 0.4650 | 0.8953 | 0.0231 | 0.5312 | 0.5200 | 0.8288 |
| SimNPO | 0.0172 | 0.6450 | 0.4601 | 0.9201 | 0.0315 | 0.5665 | 0.5135 | 0.8514 |
| RMU | 0.0211 | 0.6378 | 0.4645 | 0.9032 | 0.0387 | 0.5528 | 0.5150 | 0.8488 |
| TV | 0.0069 | 0.6340 | 0.4512 | **0.9810** | 0.0156 | 0.5012 | 0.4366 | 0.8810 |
| MOX ($\alpha = 0.5$) | 0.0146 | 0.6305 | 0.4812 | **0.9810** | 0.0163 | 0.5002 | 0.4512 | **0.9200** |
| MOX ($\alpha = 1.0$) | 0.0182 | 0.6358 | 0.4732 | 0.9788 | 0.0180 | 0.5026 | 0.4366 | **0.9120** |
| MOX ($\alpha = 2.0$) | 0.0256 | 0.6410 | 0.4555 | 0.9701 | 0.0364 | 0.5012 | 0.4330 | 0.8928 |
| MOX ($\alpha = 4.0$) | 0.0625 | **0.6504** | 0.4697 | 0.9653 | **0.0582** | **0.5219** | 0.3138 | 0.8810 |
| MOX ($\alpha = 8.0$) | 0.0319 | 0.6420 | 0.4658 | 0.9016 | 0.0340 | 0.5150 | 0.3436 | 0.8562 |
| MOX (targeted) | **0.0677** | 0.6412 | 0.4788 | 0.9710 | 0.0328 | 0.5012 | 0.3366 | 0.8858 |
| MOX (momentum) | **0.0680** | **0.6528** | **0.4410** | **0.9802** | **0.0598** | **0.5510** | **0.3120** | 0.8988 |

**Evaluation Details.** To evaluate the MU methods, we leverage TOFU (Maini et al., 2024) and MUSE (Shi et al., 2024) benchmarks. For TOFU, it contains 200 fictitious author profiles with each containing 20 question-answer pairs. To simulate the knowledge scale, there are four types of datasets in TOFU, namely Forget Set, Retain Set, Real Authors, and Real World. For evaluation on different sizes of Forget Set, TOFU provided forget ratios of $1\%$, $5\%$, and $10\%$. The evaluation metrics from TOFU mainly include Forget Quality (FQ) and Model Utility (MU). Moreover, we use ROUGE-L (Lin, 2004) on both the Forget Set and Retain Set, namely F-RL and R-RL, respectively. For MUSE, we use the New corpus that contains BBC news collected after August 2023. By following the benchmark, we evaluate four metrics, namely VerbMem on Forget Set, KnowMem on Forget Set, KnowMem on Retain Set, and PrivLeak, which denotes no verbatim memorization, no knowledge memorization, utility preservation, and no privacy leakage. All metrics are indicated with "$\uparrow$" or "$\downarrow$" to show whether a larger or smaller value leads to better performance.

**Experimental Setup** We use two LLMs, namely Llama2-7B (Touvron et al., 2023) and Phi-1.5B (Abdin et al., 2024). We use AdamW optimizer with a weight decay of 0.01 and a learning rate of $1e-5$ for training. We use a batch size of 32 and conduct 10 epochs of unlearning training for all experiments. By following Zhang et al. (Zhang et al., 2024), we use linear warm-up for the learning rate in the first epoch and then linearly decay the learning rate for the rest of the training. In the following experiments, if not specified, we choose $\alpha = 4$ with $\eta = 0.675$ for our MOX. All experiments are conducted on two H100 GPUs.

## 4.1 PERFORMANCE EVALUATION

We conduct experimental comparisons between MOX with various settings and a series of strong MU baseline methods. The results on TOFU and MUSE benchmarks are shown in Tables 1 and 2, respectively. We can see that MOX achieves effectiveness in most scenarios compared with all baseline methods. Specifically, we observe three qualities of MOX: 1) Effective forgetting, 2) Outstanding knowledge preserving, and 3) Enhanced privacy leakage.

In both tables, we can see that MOX with momentum extrapolation achieves the best performance, as it ensembles historical knowledge on the fly. But, as we tune the $\alpha$ value, we can achieve similar effectiveness as the momentum ensemble. For example, on TOFU with Llama2-7B, we can achieve comparable FQ performance when $\alpha$ increases to 4, and on MUSE, comparable FQ performance can be obtained with $\alpha = 8$. Which denotes that proper values of $\alpha$ can control the forgetting performance to reach the best result. Moreover, we find that MOX is effective in preserving the model

Table 2: Comparison of MOX and baseline methods on MUSE benchmark. The top two best performances in each column are highlighted in **bold**.

| Base LLM | Llama2-7B | | | |
|---|---|---|---|---|
| **Metric** | No Verbatim Mem. VerbMem on $\mathcal{D}_F$ ($\downarrow$) | No Knowledge Mem. KnowMem on $\mathcal{D}_F$ ($\downarrow$) | Utility Preserv. KnowMem on $\mathcal{D}_R$ ($\uparrow$) | No Privacy Leak. PrivLeak ($\in [-5\%, 5\%]$) |
| Original LLM | 58.4 | 63.9 | 55.2 | -99.8 |
| Retained LLM | 20.8 | 33.1 | 55.0 | 0.0 |
| GA | **0.0** | **0.0** | 0.0 | **17.0** |
| KL | 27.4 | 50.2 | 44.8 | -96.1 |
| GAD | 4.9 | 31.0 | 27.3 | 108.1 |
| PO | 2.3 | 21.8 | 16.1 | 109.6 |
| NPO | **0.0** | **0.0** | 0.0 | 24.4 |
| TV | 57.2 | 66.2 | **55.8** | -99.8 |
| WHP | 19.7 | 21.2 | 28.3 | 109.6 |
| MOX ($\alpha = 0.5$) | 36.5 | 38.6 | **56.2** | -93.1 |
| MOX ($\alpha = 1.0$) | 27.7 | 29.4 | 55.6 | -58.8 |
| MOX ($\alpha = 2.0$) | 18.2 | 19.8 | 55.2 | -32.0 |
| MOX ($\alpha = 4.0$) | 1.2 | 1.6 | 54.9 | -19.8 |
| MOX ($\alpha = 8.0$) | 0.8 | 1.1 | 49.5 | 35.8 |
| MOX (targeted) | **0.2** | 1.5 | 53.6 | 26.0 |
| MOX (momentum) | **0.2** | **0.8** | 54.8 | **-18.4** |

utility under most $\alpha$ values. For example, on MUSE, we observe the Utility Preserv. performance of MOX consistently surpasses most of the baseline methods, and it stays effective in most cases. However, when $\alpha$ achieves 8, such an extreme value slightly degrades the knowledge preservation in the Retain Set, and yet it still outperforms most of the baselines. At last, we find that MOX can avoid privacy leakage by tuning $\alpha$ to a proper value, as we can see that when $\alpha = 4$, the PrivLeak performance surpasses most of the baselines.

Table 3: Ablation study with various modules of MOX on TOFU dataset.

| Metric | Real Authors | | | Real World | | | Retain Set | | | Forget Set | | |
|---|---|---|---|---|---|---|---|---|---|---|---|---|
| | RL($\uparrow$) | P($\uparrow$) | TR($\uparrow$) | RL($\uparrow$) | P($\uparrow$) | TR($\uparrow$) | RL($\uparrow$) | P($\uparrow$) | TR($\uparrow$) | RL($\downarrow$) | P($\downarrow$) | TR($\uparrow$) |
| GA | 0.91 | 0.47 | 0.63 | 0.89 | 0.44 | 0.54 | 0.85 | 0.94 | 0.42 | 0.42 | 0.78 | 0.65 |
| MOX (GD) | 0.91 | 0.46 | 0.63 | 0.89 | 0.45 | 0.55 | 0.86 | 0.95 | 0.43 | 0.43 | 0.77 | 0.65 |
| MOX (GD+KL) | 0.92 | **0.49** | **0.64** | **0.90** | 0.46 | **0.57** | 0.87 | **0.96** | 0.45 | **0.40** | 0.75 | **0.67** |
| MOX (GD+target) | 0.92 | 0.47 | 0.63 | 0.89 | 0.45 | 0.55 | 0.86 | 0.94 | 0.43 | **0.40** | 0.75 | 0.66 |
| MOX (GD+KL+target) | **0.93** | **0.49** | **0.64** | **0.90** | **0.47** | **0.57** | **0.88** | **0.96** | **0.46** | 0.41 | **0.74** | 0.66 |
| NPO | 0.92 | 0.48 | 0.64 | 0.88 | 0.45 | 0.55 | 0.85 | 0.96 | 0.45 | 0.41 | 0.75 | 0.66 |
| MOX (PO) | 0.93 | 0.48 | 0.64 | 0.89 | 0.45 | 0.56 | 0.86 | **0.97** | 0.46 | 0.40 | 0.74 | 0.67 |
| MOX (PO+KL) | **0.94** | 0.48 | **0.66** | 0.89 | 0.46 | **0.57** | 0.87 | **0.97** | **0.48** | 0.39 | **0.72** | **0.68** |
| MOX (PO+target) | 0.93 | **0.49** | 0.65 | 0.89 | 0.46 | **0.57** | 0.86 | 0.96 | 0.47 | **0.39** | 0.73 | 0.67 |
| MOX (PO+KL+target) | **0.94** | **0.49** | 0.65 | **0.90** | **0.47** | **0.57** | **0.88** | **0.97** | **0.48** | **0.39** | 0.74 | **0.68** |

## 4.2 ABLATION AND ANALYTICAL STUDY

**Ablation Study.** Further, we conduct an ablation study to break down each module of MOX and understand its effectiveness. Specifically, there are four settings: 1) GD: We only conduct GD-based memorization to conduct MOX, 2) GD+KL: We conduct memorization with the KL-divergence constraint, 3) GD+target: We conduct memorization with targeted unlearning, and 4) GD+KL+target: We use all modules to realize our MOX. Moreover, we can also use PO for the memorization objective to replace GD. The result is shown in Table 3. We can see that both targeted and untargeted MOX that contain "GD+KL" can achieve the best performance under both objective settings, which again justifies the effectiveness of our method.

**Parameter Analyses.** Further, we vary the $\alpha$ and $\eta$ parameters over four datasets from TOFU to understand the sensitivity of MOX, as shown in Figure 3. We can see that larger $\alpha$ leads to better performance on the Forget Set, but it would gradually decrease the performance on the rest datasets. However, we can see that when $\alpha$ is varied from 1 to 4, the performance degradation is non-trivial; only when it is set to an extreme value like 8, the performance drop would be obvious. Hence, in general, we can set $\alpha$ to a reasonably large value to get the maximum gain on forgetting without

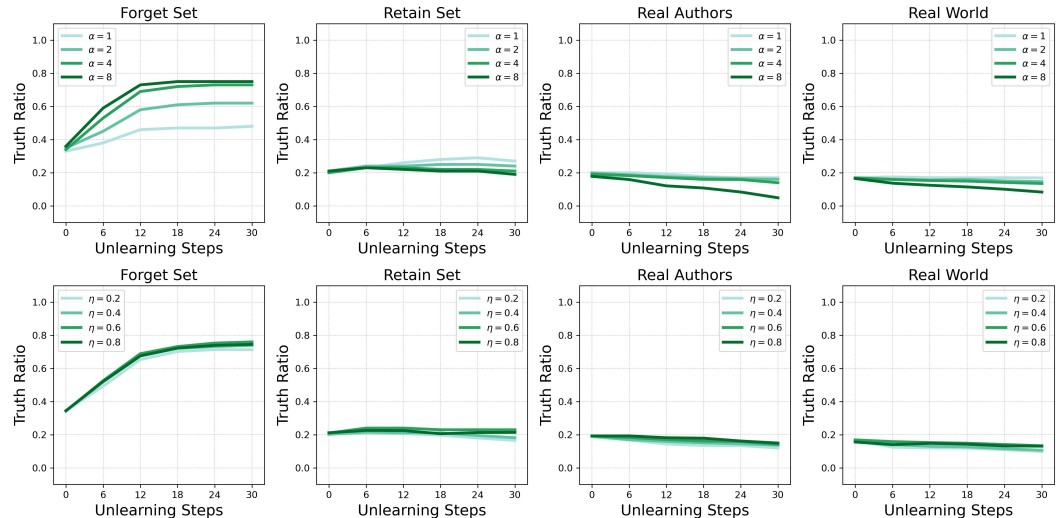

Figure 3: Parameter sensitivity analyses on $\alpha$: top row and $\eta$: bottom row.

losing much utility performance. As for $\eta$, we can clearly see that MOX performs consistently across various settings, hence, we can say MOX is actually insensitive to $\eta$.

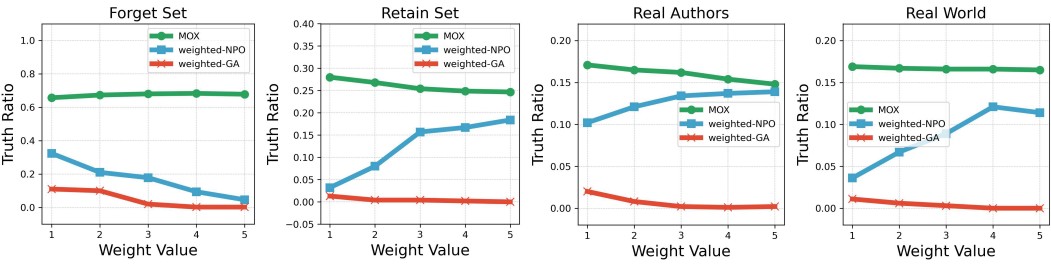

Figure 4: Performance stability under different extrapolation strengths and weight values.

**Performance Stability.** Finally, we study the performance stability of MOX across various extrapolation levels. Meanwhile, we compare MOX with GA and NPO with different regularization strengths, *i.e.*, weight value of the unlearning term. We set the weight value from 1 to 5, and show the results in Figure 4. We can see that MOX performs consistently the best among the other two methods. Moreover, it remains relatively stable under various extrapolation strengths. On the other side, the baseline methods change drastically as weight changes. Thus, we validate that MOX is stable under various settings and can be reproduced easily.

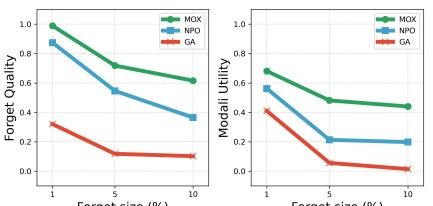

Figure 5: Performance comparison under various forget sizes.

Further, we study the stability of MOX under various sizes of forget sets, including $1\%$, $5\%$, and $10\%$. The results shown in Figure 5 demonstrate that MOX performs consistently better than both NPO and GA with a large margin, which validates the effectiveness of MOX over varied forget sizes.

## 5 CONCLUSION

In this paper, we study Machine Unlearning without GA to avoid its deficiencies. Instead of obtaining the forget model directly, we propose to first enhance memorization on the forget set. As a result, we obtain a memorization model opposite to our desired model. To achieve unlearning, we conduct model extrapolation that reaches the opposite direction from memorization. Thus, the extrapolation yields a forget model that can effectively unlearn the undesirable knowledge. Extensive evaluations justify the effectiveness and stability of our method, which shows superior forgetting performance without sacrificing model utility. Moreover, we propose targeted unlearning and momentum ensembling variants, which further enhance the adaptability and effectiveness of our method. In future studies, we hope to further understand how ensembling achieves superior performance and give a theoretical understanding of its generalization superiority.

## 6 ETHICS STATEMENT

We follow the ICLR Code of Ethics, and confirm that every author has carefully read the ICLR Code of Ethics. No violation in this paper might cause ethical issues.

## 7 REPRODUCIBILITY STATEMENT

We have carefully provided reproducible implementation details, including experimental platform, evaluation setting, baseline methods, and performance criteria. Moreover, our code is based on open-source models and codes, which do not cause any privacy issues; thus, our code will be released in the future without any copyright concerns.

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

In this appendix, we provide additional discussions and complementary analyses to further validate the performance superiority of our method MOX. First, we elucidate our implementation details. Then, we provide unlearning examples of MOX compared to other baseline methods. Further, we conduct additional empirical analyses to show the unlearning trajectories. Finally, we study the continual unlearning problem where the forget set keeps changing, and compare the performance of MOX with several typical methods.

## A ADDITIONAL DETAILS

Our experiments can be conducted on a single NVIDIA A100/H100 or 4×NVIDIA 4090 GPU. In our experiments, we use a batch size of 32. For fine-tuning pre-trained Phi-1.5B (Abdin et al., 2024), we use a learning rate of $2e-5$ for 5 epochs to obtain the original model. For Llama2-7B (Touvron et al., 2023), we also fine-tune for 5 epochs, but with a learning rate of $1e-5$. Both all models, we use AdamW as the optimizer for fine-tuning and unlearning. The unlearning process for all methods, including MOX, utilizes the same learning rate as used during fine-tuning pre-trained LLMs.

## B UNLEARNING EXAMPLE

To intuitively understand how our MOX works to unlearn specific knowledge, we conduct experiments on the Harry Potter (HP) dataset (Eldan & Russinovich, 2023). We use Llama2-7B and compare our method, MOX, with the original response. The result is shown in Table 4. We can see that the original response trained to fit the HP dataset contains lots of sensitive information, such as names from the Harry Potter book. However, the response from MOX is more general and free from the sensitive information from the HP dataset. Therefore, we can again testify that MOX can effectively unlearn sensitive knowledge and produce privacy-free content for generalized tasks.

Table 4: Unlearning examples of the original Llama2-7B and our MOX on generating responses on the HP dataset.

***Prompt***: Let's see how you would complete this piece of text: The weather did not improve as the day wore on, so that at seven o'clock that evening, when Harry and Ron went down to the Quidditch pitch for practice, they were soaked through within minutes, their feet slipping and sliding on the sodden grass.

***Original response***: The rain was pouring, thundery gray. making it difficult to see the Quaffle, and Professor McGonagall had to intervene, calling off the practice early to prevent any accidents.

***MOX***: The weather did not improve as the day wore on, and when they arrived at the field, they were soaked through within minutes, their feet slipping and sliding on the sodden grass, every movement a battle against the biting wind and driving rain.

***Question***: Let's see how you would complete this piece of text: It was the best he could do, for the icy feeling on his arm not holding the cup was not the lingering chill of the water. A slimy white hand had gripped his wrist, and the creature to whom it belonged was pulling him, slowly, backward across the rock.

***Original response***: Harry tried to resist, but he was being dragged into the darkness of the lake, the last thing he saw was the horrified faces of Ron and Hermione before he was submerged under the water.

***MOX***: It was the best he could do, for the cold feeling on his arm was not the lingering chill of the water. A slimy white hand had gripped his wrist...

## C UNLEARNING TRAJECTORY

Further, we conduct experiments on the TOFU (Maini et al., 2024) real-world dataset with $5\%$ forget set using Llama2-7B to evaluate the forget quality and model utility performance of various baseline methods, as shown in Figure 6. We can observe that several baseline methods, such as GA, GAD,

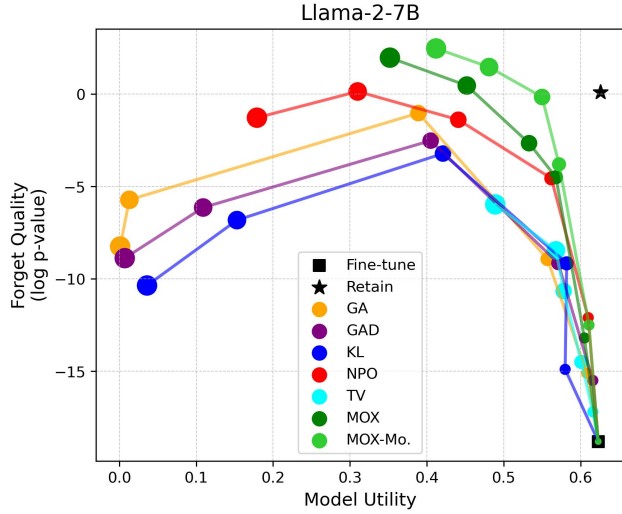

Figure 6: Learning trajectory of various unlearning methods on TOFU dataset with 5% forget set. The relative size of the markers denotes the number of epochs during unlearning.

and KL, first have an increase in their forget quality, but suddenly show a significant model utility drop, and further lead to the collapse of the forget quality. As for NPO, we can see that it is more stable compared to the above baselines, but it still shows forget quality degradation at the end of training. Most importantly, all of the baseline methods have serious model utility loss, except the TV method. Because TV also avoids gradient ascent as our MOX does, the training shows no collapse, and the forget quality is increasing for most training stages. Nonetheless, the TV method suffers from limited forget quality performance, which is far from the retained baseline model. Compared to TV, our MOX conducts more effective forgetting by enhancing the extrapolation strength. As a result, MOX can achieve significant performance improvement and still maintain satisfactory model utility results. Moreover, we observe that the momentum updated MOX, *i.e.*, MOX-Mo., can further enhance the unlearning performance on both forget quality and model utility, which again demonstrates the effectiveness of our design choices.

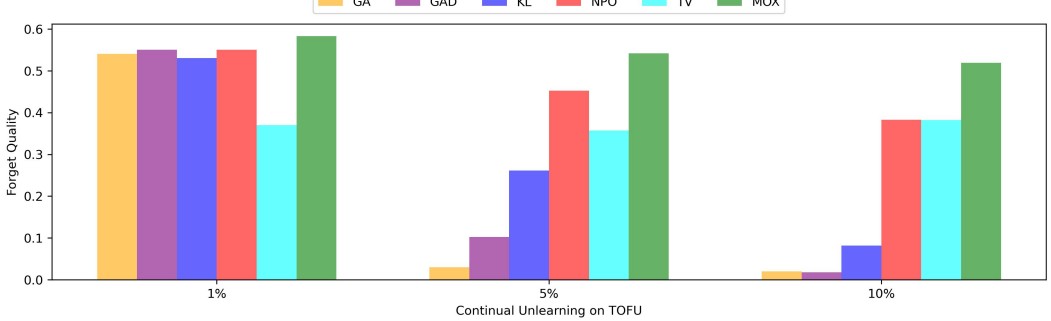

Figure 7: Continual learning performance on TOFU dataset.

## D  CONTINUAL UNLEARNING

Moreover, we study a realistic setting where the forgetting dataset keeps changing during training. We use the TOFU dataset, which contains three different forget sets with varied forget ratios, namely, 1%, 5%, and 10%. We first unlearn using the 1% forget set, then unlearn on the 5% forget set, and further the 10% forget set. For each unlearning process, we fine-tune the model for 3 epochs, and after which we evaluate the forget quality on the real-world set. The result is shown in Figure 7. We can see that some baseline methods, such as GA, GAD, and KL, significantly degrade after training

on a different forget set. Among them, we also observe that adding a retaining regularization on the retain set, as GAD and KL do, helps enhance the continual unlearning performance and prevent degradation. Moreover, the TV method is resistant to the change of the unlearning target because it avoids gradient ascent, which could be the reason to cause the instability of continual unlearning. Nonetheless, our MOX shows the most resistant performance against changing forget set and shows consistent performance superiority over all other baseline methods under all settings. Therefore, we can justify the effectiveness of MOX on continual unlearning, which brings a practical advantage for real-world applications.

| Metric | FQ (↑) | MU (↑) | F-RL (↓) | R-RL (↑) |
|---|---|---|---|---|
| GA | 0.0137 | 0.5745 | 0.4856 | 0.8795 |
| PO | 0.0501 | 0.6232 | 0.4620 | 0.8755 |
| MOX | **0.0611** | **0.6488** | **0.4500** | **0.9508** |

Table 5: Performance comparison under extreme semantic overlap between forget set and retain set.

## E  UNLEARNING WITH SEMANTIC OVERLAP

To further validate the performance of MOX when the forgetting set and retaining set are extremely overlapped, *i.e.,* share the same knowledge, we conduct experiments under TOFU by splitting the retain set into a 10% forget set and the remaining 90% as the retain set. The result is shown in Table 5. As we can see, our method is effective under strong overlapped retain and forget sets, which justifies the retention performance of MOX.

| Model | WMDP (↓) |
|---|---|
| Original | 5.21 |
| GA unlearn | 1.53 |
| GA relearn | 4.88 (+3.35) |
| NPO unlearn | 0.98 |
| NPO relearn | 5.01 (+4.03) |
| MOX unlearn | **0.54** |
| MOX relearn | **3.82** (**+3.28**) |

Table 6: Relearning performance comparison.

## F  ANALYSIS ON RELEARNING ATTACK

Relearning (Hu et al., 2024) is a popular way to recover the unlearning progress, which might be detrimental to our target. Thus, to evaluate the performance of MOX under relearning attacks, we conduct experiments to verify the robustness of MOX. We consider the WMDP benchmark using Llama2-7B, and follow the same setting as Hu et al. (Hu et al., 2024) to show the relearning performance of MOX in Table 6. We find out that the relearning performance of MOX is much more resistant than other baselines, this is because extrapolation identifies the difference between the current model and the original model, and further enhances such a difference to achieve forgetting. Hence, relearning can easily recover the effect of existing approaches, but MOX can still be resistant to the recovery during relearning.

