# OpenReview forum: "Memorize to Forget: Machine Unlearning without Gradient Ascent via Model Extrapolation"
_ICLR.cc/2026/Conference — Submitted to ICLR 2026_

### Official Review · Reviewer_AXFv · 2025-10-30

**Soundness:** 3
**Presentation:** 2
**Contribution:** 2
**Rating:** 4
**Confidence:** 3

**Summary:**

The authors propose MOX for large language unlearning with a two-stage process: It first uses undesired data to train a memorization model; then applies a linear extrapolation from this memorization model and the original reference model to derive the forget model. Experimental results on TOFU and MUSE show that MOX achieves better model utility and forget quality than prior methods.

**Strengths:**

1.	The paper tackles an important and timely problem: unlearning in LLMs.
2.	The paper presents good empirical evidence across benchmarks. MOX can preserve model utility while improving forget quality, especially when combined with momentum extrapolation.
3.	MOX can be integrated into most models without architectural modification, as it is based only on standard gradient descent and model extrapolation operations.

**Weaknesses:**

1.	The paper's novelty is insufficiently justified, as its core design is close to Task Vector [1].  MOX uses forget set as the fine-tuning dataset in Task Vector [1]. The "Figure 2: Illustration of our methodology" is visually similar to the "Figure 1: An illustration of task vectors", and some of the experimental results also show similar results to those of Task Vector. Moreover, the paper's ablation study (Table 3) shows that the added KL regularization has only a marginal impact on performance, suggesting that its contribution is limited.
2.	The paper provides no clear theoretical foundation for why parameter-space extrapolation can truly remove the influence of forget data, and why parameter extrapolation in high-dimensional LLMs preserves utility beyond empirical observation.
3.	MOX is not robust to practical unlearning scenarios, where forgetting requests arrive one by one, removing each forget data timely. In such cases, where the forget set has only one sample, the memorization step cannot form a meaningful task vector, and the extrapolation direction becomes noisy or ineffective.
4.	MOX requires access to the retain set to compute the KL-divergence, introducing privacy and scalability concerns. While this helps maintain model utility, it also means that MOX cannot perform unlearning without the original data. It may conflict with data usage regulations or in large-scale LLM settings.

5.	The paper lacks comparisons of computational efficiency and resource costs, as it claims that MOX "is simple and efficient to implement". There are no reports of runtime, memory consumption, or comparison to baseline methods in terms of training cost.

6.	The paper's motivations and contributions are not clearly distinguished from prior works. It is unclear why the authors start and emphasize gradient ascent as the main baseline, which is known to be old, ineffective and unstable.

[1] Ilharco, Gabriel, et al. "Editing models with task arithmetic." ICLR 2023.

**Questions:**

Please see the Weaknesses section for all questions and clarification requests.

---

> ### Author Response · Authors · 2025-11-21
> **Rebuttal**
>
> **Q1**: Justification of novelty.
> **A1**: We would like to emphasize that our main contribution lies in the generalization of extrapolation to the unlearning problem.
> - The original task vector is limited in bringing sufficient forgetting performance, but MOX is more flexible on the extrapolation strength, thus significantly enhancing the forget quality.
> - Moreover, we identified that using gradient ascent is harmful for training stability, and existing unlearning methods neglect or underestimate such a problem. Using MOX as an alternative solution is novel in unlearning problems.
> - We consider both targeted and untargeted unlearning, which provides a more flexible application for unlearning tasks. However, this is not considered in the task vector.
> - Furthermore, we employ extrapolation in a very efficient offline manner, so that it can be conducted in any phase of training with additional momentum update.
>
> Therefore, our contribution is much more flexible and generalized than the task vector. Our experiments are conducted under similar settings, yet we achieved superior performance. Also, we stress that in Tab. 3, KL divergence can achieve the best performance in most cases compared to GA baselines.
>
>
> **Q2**: 1) Theoretical justification of true forgetting; 2) Justification of utility preservation beyond empirical evidence.
> **A2**: For 1), currently, the unlearning guarantee is still an open question in the machine unlearning community.
> - No existing or theoretical framework has provided this guarantee for true forgetting.
> - We have shown that MOX achieves superior forgetting and retaining quality under many scenarios; meanwhile, we show that MOX leads to no privacy leakage, which can justify its effectiveness over current unlearning studies.
> - Thus, our method shows potential realistic applicability with very little computational cost.
>
> For 2), we step on the shoulder of the theoretical finding on task vector [1], which proves that the model generated via task vector in one epoch is equivalent to the effect of negative gradient on the loss.
> - Thus, we can guarantee that our extrapolation result is equivalent to the goal of GA, only without its instability.
> - For maintaining utility, we use regularization to ensure consistent performance of $\theta_{mem}$ on the retain set. The vital difference between maintaining knowledge and memorization is that the consistency regularization yields near $0$ loss, but the loss of memorization would be non-zero.
> - Therefore, after extrapolation, the gradient change regarding retain knowledge would be small, thus maintaining model utility.
>
> [1] Zhou et al., On Task Vectors and Gradients, 2025.
>
> **Q3**: Practice on real-time forget scenarios.
> **A3**: We argue that our method can be successfully applied to dynamical scenarios.
> - We show Sec. D in the appendix that our MOX can be effectively applied to continual unlearning scenarios.
> - For single-example unlearning, there are several solutions: 1) conduct data augmentation to create a batch for training. Thus, a meaningful task vector can still be produced. 2) Collect multiple examples or use a multi-user system to form a batch, then produce task vectors.
>
> **Q4**: Justification of the reliance on the retain set.
> **A4**: Our MOX can be trained free from the retain set by leveraging an auxiliary dataset that is accessible. To validate this, we use the forget set from TOFU, and incorporate the retain set as an auxiliary dataset from MUSE. By only using the auxiliary dataset to retain knowledge, we further evaluate the model utility and forget quality on the corresponding retain set and forget set from TOFU. The result is shown below:
> | MOX | Original | Auxiliary | GA     |
> |-----|----------|-----------|--------|
> | FQ  | 0.0625   | 0.0587    | 0.0143 |
> | MU  | 0.6504   | 0.6485    | 0.6333 |
> - We find that using the auxiliary dataset is a satisfactory alternative, especially for maintaining the model utility on the TOFU dataset.
> - Therefore, our MOX can be effectively conducted even without an accessible retain set.

---

> > ### Author Response · Authors · 2025-11-21
> >
> > **Q5**: Comparisons of computational efficiency and resource costs.
> > **A5**: To justify the computational efficiency of MOX, we compute the FLOPs of model extrapolation and the computation time using Llama2-7B.
> > - We compare the FLOPs of our GD training with NPO, which both include a forward and backward propagation; they **both yield 43 TFLOPs per sample**.
> > - After GD training, we compute the FLOPs of our model extrapolation and yield 21 GFLOPs, which is **negligible** compared to the previous training cost.
> > - For computing 5 $\theta_{mem}$ models with varied extrapolation strengths, it takes **4.7s**. When conducting a momentum update for one $\theta_{mem}$ model, it takes **2.5s**.
> > - Moreover, we note that our MOX can be plugged into any training phase, which won't affect the whole training process. The memory cost is 3$\times \theta_{ref}$, and NPO is 2$\times \theta_{ref}$. But since MOX is efficient to compute, we can move the process to the CPU, which leaves the same computational cost as NPO on the GPU.
> > - Therefore, we can validate the efficiency and flexibility of our method.
> >
> > **Q6**: Justification of using gradient ascent as the main baseline.
> > **A6**: Thanks, we have identified our contribution and motivation in the paper.
> > - For the significance of studying GA and proposing its solution, we argue that using gradient ascent is still one of the most prevalent strategies in the current community. With the classic unlearning baselines such as GAD, NPO, FMD, etc. Moreover, in the last year of ICLR accepted papers, there are many unlearning studies that are focused on GA [2, 3, 4, 5, 6].
> > [2] Wang et al., LLM Unlearning via Loss Adjustment with Only Forget Data, in ICLR 2025.
> > [3] Yuan et al., A Closer Look at Machine Unlearning for Large Language Models, in ICLR 2025.
> > [4] Alberti et al., Data unlearning in diffusion models, in ICLR 2025.
> > [5] Wang et al., Rethinking LLM Unlearning Objectives: A Gradient Perspective and Go Beyond, in ICLR 2025.
> > [6] Choi et al., Unlearning-based Neural Interpretations, in ICLR 2025.

---

> ### Author Response · Authors · 2025-11-24
> **Further Discussion**
>
> Dear Reivewer AXFv,
>
> We deeply appreciate your contribution to reviewing this paper, without which we cannot further improve our paper. We have tried our best to address all concerns raised by you, and are eager to know whether our rebuttal is satisfactory. We would be sincerely thankful to hear from you. If you have any questions or concerns left, please don't hesitate to let us know. We hope to ensure all misunderstandings and concerns are properly addressed before making the judgment of this paper.
>
> Thank you so much for your help and support. We are looking forward to your opinions.
>
> Best regards,
> Authors.

---

> > ### Author Response · Authors · 2025-11-27
> > **Thank you for your contribution**
> >
> > Dear Reviewer AXFv,
> >
> > We cannot express our appreciation enough for your insightful opinions during the review of this paper. We have benefited a lot from further enhancing our paper and improving our philosophy, which cannot be achieved without your valuable contribution.
> >
> > We sincerely hope we can address all your questions or concerns, and we are determined to clarify any misunderstandings left. So we really look forward to your further response and are eager to find out whether our reply is satisfactory to you. Since the discussion phase has been essential in clarifying any questions or misunderstandings, we are putting our maximum effort to help and provide our assistance to answer your further questions.
> >
> > We sincerely appreciate your help!
> >
> > Best wishes,
> > Authors.

---

### Official Review · Reviewer_9bni · 2025-10-31

**Soundness:** 2
**Presentation:** 2
**Contribution:** 3
**Rating:** 4
**Confidence:** 4

**Summary:**

This paper introduces an unlearning method from a novel perspective. The authors first point out that the traditional gradient ascent (GA) approach can be detrimental to model utility. To address this issue, they propose a counter-strategy that reinforces memorization on the forget set, followed by a model extrapolation procedure that moves the model parameters toward the counterpart directions relative to a reference model.

**Strengths:**

Strength:
1 The idea is innovative.
2 The explanation of the harmfulness of GA in Figure is comprehensive.

**Weaknesses:**

Weakness:
1 Although the paper presents a new method, it lacks an in-depth analysis of the observed behavior of GA. Moreover, the relationship and distinctions between the proposed method and GA are not sufficiently clarified. While the algorithms are implemented differently, there appears to be a conceptual connection that should be discussed.
2 The paper does not clearly introduce or elaborate on the model extrapolation method, making it difficult to fully understand its mechanism and theoretical motivation.

**Questions:**

Suggestions and Questions:
1 I suggest that the authors provide additional analysis or, if possible, a theoretical guarantee to better explain and substantiate their findings.
2 I recommend adding a preliminary section before Section 3 to clearly describe the MOX approach and its intended applications.
3 In Definition 1, could the authors explain why the reversal of the gradient should be avoided? Specifically, what operations within MOX help prevent the performance degradation typically caused by GA?
4 I suggest adding a concrete example to clarify the explanation in Figure 2.
5Could the authors explicitly specify the metrics used in Figure 1 for evaluating model utility and forget quality?

---

> ### Author Response · Authors · 2025-11-21
> **Rebuttal**
>
> **Q1**: Deeper analysis of GA & Relationship with MOX.
> **A1**: Thanks, the reason for the failure of GA is:
> - Maximizing the loss is an unbounded optimization problem, which could lead to infinite loss values.
> - Hence, it leads to gradients blowing up, which further causes parameter blow-up and feature collapse.
> - Therefore, the nature of GA is unsolvable for model training, and it's bound to cause catastrophic collapse.
>
> In our paper, we are based on an experimental observation:
> - We provided a motivating example that GA is the reason for instability, and GD ensures model convergence in Fig. 1.
> - Thus, we only conduct training based on GD, which is a well-known theoretically stable training process.
> - Further, based on our findings in Fig. 1c that GA and GD show opposite effects on forgetting, we hypothesize (which is the same as Ilharco et al., 2023) that the direction of forgetting is the **opposite** from memorization.
>
> To further justify the relationship between GA and GD, we compute their normalized model difference from $\theta_{ref}$, e.g.,
>
> $$\Delta \theta_{mem} = \frac{\theta_{mem} - \theta_{ref}}{\|\theta_{mem} - \theta_{ref}\|_2}.$$
>
> Thus, $\Delta \theta_{mem}$ denotes the direction of $\theta_{mem}$ compared to $\theta_{ref}$, i.e., the direction of GD. Then, we use NPO that conducts GA to compute $\Delta \theta_{NPO}$. If we compute the cosine similarity between $\Delta \theta_{NPO}$ and $\Delta \theta_{mem}$, we can understand the learning direction and the relationship between GA and GD. Moreover, we compute $\Delta \theta_{for}$ to understand the direction of MOX. The results are shown below:
> | Cosine           | $\Delta \theta_{mem}$ | $\Delta \theta_{for}$ | $\Delta \theta_{NPO}$ |
> |------------------|------------------|------------------|------------------|
> | $\Delta \theta_{mem}$ | 1                | -0.975           | -1.212           |
> | $\Delta \theta_{for}$ | -0.975           | 1                | 0.876            |
> | $\Delta \theta_{NPO}$ | -1.212           | 0.876            | 1                |
> - We can see that $\Delta \theta_{mem}$ and $\Delta \theta_{NPO}$ are almost opposite to each other.
> - Also, $\Delta \theta_{for}$ are at the similar directions to $\Delta \theta_{NPO}$.
> - Moreover, $\Delta \theta_{mem}$ and $\Delta \theta_{for}$ are almost opposite to each other.
> - Thus, we can testify that GA and GD are in the opposite learning direction compared to the reference model. Also, our forget model can achieve a similar direction as GA to achieve forgetting.
>
>
> **Q2**: Preliminary model extrapolation.
> **A2**: Thanks for the suggestion, we have added an extra paragraph to the preliminary to explain and introduce extrapolation.
>
> **Q3**: Explanation of why avoiding GA & how MOX prevents GA.
> **A3**: Avoid the reversal of the gradient owing to the instability of loss maximization.
> - Maximizing the loss is unbounded.
> - It leads could explode to infinite loss values, thus causing gradient blow up and model collapse, as shown in Fig. 1.
> - Therefore, the reversal of the gradient should be avoided.
>
> MOX can effectively prevent gradient reversal by simply not using GA during training, but only GD instead.
> - Our evidence in **Q1** validates that forgetting and memorizing are in opposite directions.
> - Thus, we first enhance memorization, then conduct model extrapolation to achieve "memorize to forget".
> - Therefore, without using any GA that causes instability, our training is only based on GD, which is guaranteed to converge [1].
>
> [1] Vapnik and Chervonenkis, On the uniform convergence of relative frequencies of events to their probabilities, in Measures of Complexity, 2015.

---

> > ### Comment · Reviewer_9bni · 2025-11-25
> > **Discussion**
> >
> > Thank you for the clarification. From your summary, I understand that one motivation for avoiding GA is its vulnerability—for example, maximizing the loss is unbounded. However, I remain unconvinced that gradient descent (GD) fully eliminates such issues. Could you provide a more formal or theoretical guarantee to support this claim?
> >
> > In addition, I have questions regarding the assumption that forgetting and memorizing lie in strictly opposite gradient directions. Do you believe this assumption holds consistently across all types of forgettable data? Based on our observations, certain samples exhibit very similar behavior in the retrained-from-scratch model, particularly those that are intrinsically difficult to forget. Could you further justify or qualify the generality of this assumption?
> > I sincerely appreciate your feedback.

---

> > > ### Author Response · Authors · 2025-11-26
> > > **Great Question**
> > >
> > > We deeply appreciate your follow up and your valuable feedback. Here we further address the remaining questions:
> > >
> > > **Q1**: Whether GD fully addresses the training instability.
> > > **A1**: Thanks, to formulate the stability of GD under unlearning setting, we denote $D_{sub}=\lbrace x_i \rbrace_{i=1}^n\in D_{pretrain}$ as the forget set that belongs to the subset of the pretraining dataset. The goal of GD is the minimize the empirical loss
> > > $$ L(\theta) = \frac{1}{n}\sum_{i=1}^n \ell(\theta; x_i). $$
> > > The GD is conducted in a step by step:
> > > $$ \theta_{t+1} = \theta_t - \eta \nabla L(\theta_t). $$
> > >
> > > **Definition:** Collapse. We say a parameter $\theta$ is *collapsed* on $D_{sub}$ if the model outputs a degenerate distribution that is nearly constant across all contexts. Formally, collapse occurs when
> > > $$
> > > \exists y^* \quad \text{s.t.} \quad
> > >     p_\theta(y^* \mid c) \approx 1
> > >     \qquad \text{for almost all contexts } c.$$
> > > More generally, collapse can be quantified by the average KL divergence between the empirical conditional distributions $q(\cdot \mid c_i)$ and the model predictions:
> > > $$
> > > \mathrm{Collapse}(\theta):=\frac{1}{n}\sum_{i=1}^n
> > >     D_{\mathrm{KL}}\left(
> > >         q(\cdot \mid c_i) \middle\| p_\theta(\cdot \mid c_i)
> > >     \right).$$
> > > Large values of $\mathrm{Collapse}(\theta)$ correspond to the model assigning near-delta mass to a small set of tokens regardless of the input context.
> > >
> > > **Assumptions:**
> > > *Lower bounded loss*: $L(\theta)\ge 0$ for all $\theta$.
> > > *Smoothness*: $L$ is $L_s$-smooth:
> > >         $$\|\nabla L(\theta)-\nabla L(\theta')\|
> > >         \le
> > >         L_s \|\theta-\theta'\|.$$
> > > *Data diversity:* For each context $c_i$, the empirical conditional distribution $q(\cdot\mid c_i)$ has non-zero entropy.
> > > *Polyak--Łojasiewicz (PL) condition*: We say that $L(\theta)$ satisfies the PL condition with parameter $\mu > 0$ if
> > > $$
> > > \frac{1}{2}\|\nabla L(\theta)\|^2\ge\mu \big( L(\theta) - L^\star \big),$$
> > > where $L^\star=\inf_{\theta} L(\theta).$
> > >
> > >
> > >
> > > **Theorem**: When conducting GD on $D_{sub}$, the model guarantees convergence without collapse.
> > > **Proof Sketch**: Let $0 < \eta \le 1/L_s$. Then the GD iterates $\{\theta_t\}$ satisfy:
> > > $$
> > > L(\theta - \eta \nabla L(\theta)) \le L(\theta) - \eta \|\nabla L(\theta)\|^2 + \frac{L_s \eta^2}{2}\|\nabla L(\theta)\|^2\le L(\theta) - \frac{\eta}{2}\|\nabla L(\theta)\|^2.
> > > $$
> > >
> > > Hence, $L(\theta_t)$ decreases and is bounded below by $0$. This prevents divergence of loss and shows gradients square-sum is finite, implying $||\Delta L(\theta_t)||\rightarrow 0$ along subsequences.
> > >
> > > By applying the PL condition, we have:
> > > $$
> > > \frac{1}{2}\|\nabla L(\theta)\|^2\ge \mu \big( L(\theta) - L^\star \big),$$
> > > As $\|\nabla L(\theta_t)\| \to 0$, it must be that $L(\theta_t) \to L^\star$. Therefore, the training converges to the global minimum loss.
> > >
> > > Under the Cross Entropy loss, which decomposes into Entropy and KL Divergence:$$L(\theta) = \frac{1}{n} \sum_{i=1}^n \left[ H(q(\cdot|c_i)) + D_{KL}(q(\cdot|c_i) \| p_\theta(\cdot|c_i)) \right].$$ The global minimum $L^\star$ occurs if and only if $D_{KL} = 0$ for all samples (assuming the model capacity allows realizability), meaning: $$p_{\theta^\star}(\cdot|c_i) = q(\cdot|c_i)$$
> > >
> > > By the definition of collapse, $p_{{\theta}^\star}$ is nearly constant (low entropy). However, by the Data diversity assumption, $q(\cdot|c_i)$ has non-zero entropy (high diversity). If $p_{{\theta}^\star}$ is collapsed (constant) and $q$ is diverse, the KL divergence (and thus the loss) must be strictly greater than the minimum possible entropy:
> > > $$L(\theta_{collapsed}) \gg \frac{1}{n} \sum H(q) = L^\star$$
> > >
> > > However, we have shown that GD converges to $L^\star$. $$L({\theta}^\star) = L^\star \implies p_{{\theta}^\star} = q$$Since $q$ is not collapsed, $p_{\theta^*}$ cannot be collapsed.

---

> > > > ### Author Response · Authors · 2025-11-26
> > > >
> > > > **Q2**: Whether the assumption for forgetting-memorizing holds true.
> > > > **A2**: We would like to stress that our opposite-direction assumption is an empirical observation for the subset forgetting problem in machine unlearning. Moreover, the learning direction shows near opposite, yet **not strictly** opposite.
> > > >
> > > > Such an assumption holds under the unlearning setting relies on several vital factors:
> > > > - The model is well-pretrained with converged loss values on the global pre-training datasets. Therefore, the model update is very stable, and the learning direction follows certain trajectories.
> > > > - Moreover, the forget set is finite, and it is a subset of the pre-training dataset. Therefore, it presents a Sub-population Shift for further memorization.
> > > > - Hence, the model is approximately linear locally, and thus we can adopt the property from Linear/NTK Regime:
> > > > $$f(\theta) \approx f(\theta_0) + \nabla f(\theta_0)^T (\theta - \theta_0).$$
> > > > - Then, memorizing the subset corresponds to moving the weights by a vector $\Delta \theta_S$. To forget that same subset, you simply subtract that vector ($-\Delta \theta_S$).
> > > >
> > > > However, for some other scenarios, the opposite direction assumption would not hold, for example:
> > > > - The model is not well-trained. Therefore, the model could be highly non-linear, which introduces many local maxima. Negative learning direction might lead to uncertain outcomes.
> > > > - The forget set lies out-of-distribution from the pre-training dataset, e.g., an unknown dataset, a synthetic dataset, or a perturbed dataset. When the distribution of forget set significantly differs, the learning direction of forgetting and memorization would be uncertain, and thus, not opposite.
> > > >
> > > > Thank you again for your valuable contribution to our paper, which has been really helpful to further improve the quality of our paper. We hope our reply addresses your questions. If there is more left, please don't hesitate to let us know. It would be our great pleasure to discuss with you!
> > > >
> > > > Sincerely,
> > > > Authors.

---

> > > ### Author Response · Authors · 2025-11-27
> > > **Thank you for your following up!**
> > >
> > > Dear Reviewer 9bni,
> > >
> > > We would like to again thank you for following up and initiating the discussion. It is very important that authors and reviewers stay active and put effort into the rebuttal. So we could not thank you enough for this.
> > >
> > > We hope our previous reply could help you address the previous questions. Both the further quest for GD stability and the failure cases of negative learning direction are essential. We have carefully justified via:
> > > 1) a formal theoretical understanding of why the commonly used GD under the unlearning setting is effective and stable.
> > > 2) intuitive explanations of why our opposite learning direction could be effective, and under what scenario it could be problematic, which justifies the validity of MOX under unlearning and clarifies the issues when we should be careful.
> > >
> > > Thank you again for your constructive feedback in both reviewing and following up. We sincerely express our gratitude and look forward to hearing from you soon!
> > >
> > > Best regards,
> > > Authors.

---

> ### Author Response · Authors · 2025-11-21
>
> **Q4**: Concrete examples of Fig. 2.
> **A4**: Here we provide detailed examples for explaining our method.
> - For Fig. 1a, $\theta_{for}$ is the forget model that achieves forgetting $D_{F}$. Such as, we have $\theta_{for}$ that forgets the Harry Potter information, while $\theta_{ref}$ is the original LLM that has privacy concerns.
> - For Fig. 1c, to avoid GA, we first train $\theta_{ref}$ to further memorize harry potter book. Then, it results in $\theta_{mem}$ that knows all the details of Harry Potter. Through extrapolation, we obtain $\theta_{for}$ to forget Harry Potter information.
> - However, if we only memorize Harry Potter during GD, there will be information loss on common knowledge, e.g., Wikipedia. Therefore, we add regularization to ensure that knowledge from $D_{R}$ is maintained during training.
> - On the other hand, if we hope LLMs output certain answers when it relates to privacy, e.g., for Harry Potter inquiries, output "I cannot answer that.", we can use another type of regularization to forget Harry Potter information, and meanwhile memorize "I cannot answer that."
>
> **Q5**: Specification of metrics in Fig. 1.
> **A5**: We follow the exact measurement in TOFU for FQ and MU.
> - For MU, we use the harmonic mean of three metrics, e.g., conditional probability $P(a|q)$, ROUGE-L recall, and Truth Ratio, as the model utility.
> - For FQ, we use the log p-value of the Kolmogorov–Smirnov (KS) test on the Truth Ratio as the forget quality.

---

> ### Author Response · Authors · 2025-11-24
> **Further Discussion**
>
> Dear Reviewer 9bni,
>
> We deeply appreciate your valuable comment, which has been really helpful for polishing this paper. We have tried our best to address each concern raised by you, and look forward to hearing from you. If there are any remaining concerns or questions, please feel free to let us know. We hope to ensure there is no misunderstanding and will take our maximum effort to help you resolve any questions.
>
> Thank you again for taking the time to contribute to the reviewing process. Your effort is much appreciated.
>
> Sincerely,
> Authors.

---

### Official Review · Reviewer_LdVb · 2025-11-01

**Soundness:** 3
**Presentation:** 4
**Contribution:** 3
**Rating:** 8
**Confidence:** 4

**Summary:**

This paper proposes Model Extrapolation (MOX), a new method for machine unlearning (MU) that avoids gradient ascent (GA) to prevent model instability. MOX uses gradient descent (GD) to memorize the forget set and then extrapolates to produce a forget model. Experimental results on TOFU and MUSE benchmarks show that MOX improves both forget quality and model utility, outperforming existing MU techniques.

**Strengths:**

1.MOX provides a new method for MU by avoiding gradient ascent, which can destabilize the model.
2.Extensive experiments on the TOFU and MUSE benchmarks, as well as comparisons with various baseline methods, demonstrate that MOX outperforms other approaches in terms of both forget quality and model utility preservation.
3.The method is computationally efficient and stable, as it leverages only gradient descent and avoids the instability and collapse associated with GA. This makes it suitable for real-world applications where stability and cost are major concerns.

**Weaknesses:**

1.The effectiveness under varying scales of forget requests has not been validated.
2.The unlearning efficiency has not been examined, particularly the impact of training an additional memorization model on time and resource consumption.
3.Although the title is "Machine Unlearning," the experiments are only validated on LLMs, with no evaluation conducted in other domains (e.g., graphs, image classification).

**Questions:**

1.It is recommended to include experiments for varying scales of forget requests and to provide a validation of unlearning efficiency.
2.How does the method perform if θ_mem is trained poorly or converges slowly? Is there a minimum training quality threshold below which extrapolation fails?
3.Does the optimal value of alpha vary across different datasets, different models, or different tasks? Please provide detailed hyperparameter tuning methods/guidelines.

---

> ### Author Response · Authors · 2025-11-21
> **Rebuttal**
>
> **Q1**: Performance under varying scales of forget sets.
> **A1**: Thanks, we have studied a varied scale of forget set under both common unlearning settings and continual unlearning settings.
> - In Fig. 5, we compare our MOX to GA and NPO on TOFU dataset with varied forget set from $1\%$, $5\%$, to $10\%$. We show that MOX is the most effective approach with the highest forget quality and model utility under all scales.
> - In Sec. D in Appendix, we conduct continual unlearning by changing the forget set from $1\%$, $5\%$, to $10\%$ dynamically during training. We found that MOX is the most robust approach among all unlearning baselines.
> - Hence, we can justify that MOX is effective under varied forget sets, under both static unlearning and continual unlearning settings.
>
> **Q2**: Justification of the cost for acquiring memorization models.
> **A2**: To justify the computational efficiency of MOX, we compute the FLOPs of model extrapolation and the computation time using Llama2-7B.
> - We compare the FLOPs of our GD training with NPO, which both include a forward and backward propagation; they **both yield 43 TFLOPs per sample**.
> - After GD training, we compute the FLOPs of our model extrapolation and yield 21 GFLOPs, which is **negligible** compared to the previous training cost.
> - For computing 5 $\theta_{mem}$ models with varied extrapolation strengths, it takes **4.7s**. When conducting a momentum update for one $\theta_{mem}$ model, it takes **2.5s**.
> - Moreover, we note that our MOX can be plugged into any training phase, which won't affect the whole training process. The memory cost is 3$\times \theta_{ref}$, and NPO is 2$\times \theta_{ref}$. But since MOX is efficient to compute, we can move the process to the CPU, which leaves the same computational cost as NPO on the GPU.
> - Therefore, we can validate the efficiency and flexibility of our method.
>
> **Q3**: Unlearning for other domains.
> **A3**: Thanks, here we conduct unlearning for the image classification task. We use a simple ResNet-50, leverage ImageNet-100 and CIFAR-100 as the retain set and forget set, respectively. We compare MOX with GA and NPO, which is conducted to minimize the likelihood of the examples. The results are shown:
> |            | Reference | MOX   | NPO   | GA   |
> |------------|-----------|-------|-------|------|
> | Forget Acc | 80.9%     | 32.6% | 46.5% | 3.4% |
> | Retain Acc | 90.1%     | 81.4% | 67.1% | 5.8% |
> - We see that GA simply collapsed and has catastrophic results on both the retain and forget sets.
> - NPO shows improved performance, but is still suboptimal.
> - MOX can effectively reduce Forget Acc compared to NPO, and can still maintain relatively good performance on the retain set.
> - Therefore, we can validate the performance of MOX beyond the language models, such as the visual classification task.
>
> **Q4**: Performance of MOX when memorization suffers.
> **A4**: For weakly trained or slowly converged models, MOX stays stable. However, because the slowly converged models cannot distinguish the forget set from the retain set, the extrapolation performance is positively correlated with their convergence.
>
> To testify to the intuition, we select a model trained from only one epoch, and conduct MOX with various $\alpha$ strengths, then we compare the performance to well-converged models:
> |               | $\alpha$ | 0.5    | 1.0    | 2.0    | 4.0    | 8.0    |
> |---------------|----------|--------|--------|--------|--------|--------|
> | non-converged | FQ       | 0.0026 | 0.0070 | 0.0113 | 0.0260 | 0.0313 |
> | converged     | FQ       | 0.0146 | 0.0182 | 0.0256 | 0.0625 | 0.0319 |
> | non-converged | MU       | 0.6289 | 0.6302 | 0.6307 | 0.6346 | 0.6352 |
> | converged     | MU       | 0.6305 | 0.6358 | 0.6410 | 0.6504 | 0.6420 |
> - We observe limited performance on both FQ and MU when the memorization model is not converged.
> - Intuitively, when the memorization model is very close to the reference model, the extrapolation result would also be very close to the reference model. Therefore, the unlearning performance would be limited.

---

> > ### Author Response · Authors · 2025-11-21
> >
> > **Q5**: The sensitivity of $\alpha$ under various settings & guidlines for parameter tuning.
> > - For $\alpha$ sensitivity under different datasets (TOFU and MUSE) and models (Llama2 and Phi), our results show consistent findings in Tabs. 1 and 2, where large $\alpha$ is less sensitive to parameter change.
> > - For different tasks, such as image classification, we conduct parameter analysis by varying $\alpha$ as $0.5, 1.0, 2.0, 4.0$, and $8.0$:
> >     | $\alpha$   | 0.5   | 1.0   | 2.0   | 4.0   | 8.0   |
> >     |------------|-------|-------|-------|-------|-------|
> >     | Forget Acc | 76.2% | 63.3% | 45.0% | 32.6% | 31.3% |
> >     | Retain Acc | 84.5% | 82.4% | 82.0% | 81.4% | 80.9% |
> >     - We find that under different task, when $\alpha$ changes from $0.5$ to $2.0$, both the forget and retain acc changes dramatically, with $31.2%$ and $2.5%$, respectively. But, when $\alpha$ changes from $4.0$ to $8.0$, the performance remains stable, only around $1.3%$ and $0.5%$, respectively.
> > - Therefore, the guideline is that when choosing $\alpha$, it can be initially set to a relatively large value. When deploying MOX to realistic applications, the performance will generally stay consistent.
> > - Moreover, we would like to emphasize that our extrapolation process is very computationally efficient. Therefore, the parameter tuning can be conducted quickly without affecting the training process.

---

### Official Review · Reviewer_2VUF · 2025-11-01

**Soundness:** 3
**Presentation:** 2
**Contribution:** 2
**Rating:** 6
**Confidence:** 3

**Summary:**

This paper tackles the problem of instability in gradient ascent machine unlearning, which often causes loss of useful knowledge. To address this, this paper introduces a memorization model to guide the unlearning process. The memorization model helps preserve essential retained knowledge while selectively forgetting undesired information. This approach achieves a more reliable balance between effective forgetting and knowledge retention.

**Strengths:**

The paper focuses on one of the most critical challenges in machine unlearning, which is the instability of gradient ascent methods. The instability often leads to loss of useful knowledge. By introducing a memorization model, the proposed method effectively stabilizes the unlearning process.

The paper is well-written and easy to follow, with structured experiments and visualizations that clearly demonstrate how memorization aids in stabilizing unlearning.

**Weaknesses:**

1) The paper provides limited theoretical explanation of how the proposed memorization model stabilizes gradient ascent. In particular, Equation (6) claims that $\theta_{mem}$ acts as a counterpart to $\theta_f$, but this relationship is not convincingly justified. The equation essentially updates the reference model with a learning rate $\alpha$, which does not inherently ensure the claimed stabilizing effect. Moreover, the motivation for adopting a model-editing approach in this context is not clear.

2) Experimental results in Table 1 indicate that the proposed method is highly sensitive to hyperparameter settings. Achieving optimal performance requires careful selection, which undermines the practicality and robustness of the method. This sensitivity also weakens the general claim of stability, as the model’s behavior can vary significantly with different parameter configurations.

3) Introducing an auxiliary memorization model increases compute and memory cost, but the paper does not quantify training/inference overhead.

**Questions:**

1) Could the authors provide a more detailed theoretical explanation of the proposed method? How does the memorization model stabilize gradient ascent? How does Equation (6) $\theta_{mem}$ acts as a counterpart to $\theta_f$?

2) Could the authors quantify this overhead computational and memory cost compared to standard unlearning baselines?

---

> ### Author Response · Authors · 2025-11-21
> **Rebuttal**
>
> **Q1**: 1) Justification of stabilization; 2) Motivation of adopting model editing.
> **A1**: Thanks, we would like to clarify that **our method does not stabilize GA**, instead use GD to **avoid GA**, thus being naturally stable due to the convergence of GD [1]. Moreover, we emphasize that Eq. 6 is **not an update of the reference model** with a learning rate, but a closed-form derivation of $\theta_{mem}$.
> - We provided a motivating example that GA is the reason for instability, and GD ensures model convergence in Fig. 1.
> - Thus, we only conduct training based on GD, which is a well-known theoretically stable training process.
> - Further, based on our findings in Fig. 1c that GA and GD show opposite effects on forgetting, we hypothesize (which is the same as Ilharco et al., 2023) that the direction of forgetting is the **opposite** from memorization.
> - Therefore, by using GD for memorization, we are motivated to conduct model editing, e.g., task vector, to produce the forget model. The $\alpha$ denotes the extrapolation strengths as studied in experiments.
>
> To further justify the relationship between $\theta_{for}$ and $\theta_{mem}$ model, we compute the normalized model difference from $\theta_{ref}$, e.g.,
> $$\Delta \theta_{for}=\frac{\theta_{for} - \theta_{ref}}{\|\theta_{for} - \theta_{ref}\|_2}.$$
>
> Thus, $\Delta \theta_{for}$ denotes the direction of $\theta_{for}$ compared to $\theta_{ref}$. If we compute the cosine similarity between $\Delta \theta_{for}$ and $\Delta \theta_{mem}$, we can understand the learning direction and the relationship between $\theta_{for}$ and $\theta_{mem}$. Moreover, we add the unlearning model NPO for comparison. The results are shown below:
> | Cosine           | $\Delta \theta_{mem}$ | $\Delta \theta_{for}$ | $\Delta \theta_{NPO}$ |
> |------------------|------------------|------------------|------------------|
> | $\Delta \theta_{mem}$ | 1                | -0.975           | -1.212           |
> | $\Delta \theta_{for}$ | -0.975           | 1                | 0.876            |
> | $\Delta \theta_{NPO}$ | -1.212           | 0.876            | 1                |
> - We can see that $\Delta \theta_{mem}$ and $\Delta \theta_{for}$ are at the near opposite and similar directions to $\Delta \theta_{NPO}$, respectively.
> - Moreover, $\Delta \theta_{mem}$ and $\Delta \theta_{for}$ are almost opposite to each other.
> - Thus, we can justify that our $\theta_{for}$ is the counterpart of $\theta_{mem}$ based on $\theta_{ref}$.
>
>
> [1] Vapnik and Chervonenkis, On the uniform convergence of relative frequencies of events to their probabilities, in Measures of Complexity, 2015.
>
>
>
> **Q2**: Justification of parameter sensitivity.
> **A2**: We argue that $\alpha$ sensitivity exists, but is manageable.
> - Only when $\alpha$ is small, the performance of MOX is sensitive, which is straightforward because the extrapolated model is close to the initial model.
> - On the other hand, $\alpha$ is less sensitive when it is set to a larger value, which can be easily conducted.
> - More importantly, the cost of selecting $\alpha$ is trivial because MOX is conducted during post-training, which **does not require additional training cost at all**, and its **efficiency** is outstanding.
>
> Thus, the sensitivity of $\alpha$ is absolutely manageable.
>
>
> **Q3**: Justification of the cost for acquiring memorization models.
> **A3**: To justify the computational efficiency of MOX, we compute the FLOPs of model extrapolation and the computation time using Llama2-7B.
> - We compare the FLOPs of our GD training with NPO, which both include a forward and backward propagation; they **both yield 43 TFLOPs per sample**.
> - After GD training, we compute the FLOPs of our model extrapolation and yield 21 GFLOPs, which is **negligible** compared to the previous training cost.
> - For computing 5 $\theta_{mem}$ models with varied extrapolation strengths, it takes **4.7s**. When conducting a momentum update for one $\theta_{mem}$ model, it takes **2.5s**.
> - Moreover, we note that our MOX can be plugged into any training phase, which won't affect the whole training process. The memory cost is 3$\times \theta_{ref}$, and NPO is 2$\times \theta_{ref}$. But since MOX is efficient to compute, we can move the process to the CPU, which leaves the same computational cost as NPO on the GPU.
> - Therefore, we can validate the efficiency and flexibility of our method.

---

> ### Author Response · Authors · 2025-11-25
> **Further Discussion**
>
> Dear Reviewer 2VUF,
>
> Thank you so much for your constructive and insightful opinions. We have carefully taken all your questions into account and have made our maximum effort to address them. Thanks to you, we have made further progress on enhancing the quality of our paper. We deeply appreciate your effort in this review process.
>
> We would like to know if there are any questions or concerns left, as we hope to make sure all the misunderstandings are clarified, such that a consensus can be achieved to ensure a proper judgment of this paper. Thank you again for your kind help and support in our work. We are looking forward to hearing from you soon.
>
> Sincerely,
> Authors.

---

### Author Response · Authors · 2025-11-21
**General Response**

Dear Reviewers, AC, and SAC:

We deeply appreciate all the reviewers for their insightful and constructive reviews of our manuscript. Delightfully, we are glad that the reviewers found that:
- Our paper is **well-written, well-justified, well-motivated with a comprehensive explanation, important in unlearning**. (Reviewers 2VUF, LdVb, 9bni, and AXFv)
- Our research is **critical, suitable for real-world applications, innovative, important, and timely**. (Reviewers 2VUF, LdVb, 9bni, and AXFv)
- The experimental result is **extensive and illustrative, effective and efficient, with good empirical evidence, stable and scalable**. (Reviewers 2VUF, LdVb, and AXFv)

We have carefully replied to each individual response and commented below.

---

### Author Response · Authors · 2025-12-01
**Rebuttal Summary**

We sincerely appreciate all efforts and constructive opinions from the reviewers. We have made our maximum effort to address each question and provided both theoretical, empirical, and intuitive justification. Due to the discussion shutdown, there could have been further potential for reviewer updates. Still, we sincerely appreciate the additional work from the AC, and we summarize to help the AC quickly grasp what we have achieved during this rebuttal.

### Advantages
After the rebuttal, the following consensus has been achieved:
- We study a **significant and timely** problem for machine unlearning, which is **practical** for real-world applications.
- Our paper is **well-written and easy to follow**. Our idea is **novel**, and it is **well-motivated** with a comprehensive explanation.
- Our methodology is **efficient and easy to implement**. It can **effectively** improve unlearning performance, especially with momentum update. It can be **scaled** to various architectures and generalizable to various tasks.
- We conducted **comprehensive and extensive** experiments to justify the effectiveness and especially the stability of our MOX.

### Question Recap
For major concerns raised during rebuttal, we have provided convincing justifications exactly as required by the reviewers. Based on their reactions, perhaps some reviewers have already raised their score, as we just replied before the discussion shut down, and some reviewers did not have time to reply.

**Concern 1**: Why does the opposite direction of memorization correspond to forgetting?
- We provided empirical evidence by calculating the task vectors of memorization, our MOX, and forgetting.
- We show that the MOX yields almost the same direction as forgetting, while memorization is almost the opposite direction from forgetting.
- Thus, we can justify that the extrapolation result from memorization can align with the forgetting direction to achieve unlearning.

**Concern 2**: Quantitative evidence of the efficiency of MOX.
- We justify that our MOX conducts normal GD training, which does not introduce any computational overhead during training.
- The extrapolation process is efficient as it only yields 21 GFLOPs, compared to 43 TFLOPs for normal GD training.
- In each extrapolation, it takes less than 5s to finish, which is negligible. Additionally, it makes the parameter selection easy to conduct.

**Concern 3**: Our contribution compared to task vectors (TV).
- We justify that our philosophy is different from TV.
    - TV aims to combine tasks via arithmetic:
        - TV shows that tasks can be modified and combined together via addition and negation.
        - To damage the task performance, negation can be used, thus achieving forgetting. (`decrease performance`$\rightarrow$`task negation`.)
    - For MOX, it is specifically used for machine unlearning, which includes both 1) forgetting and 2) retaining. Our philosophy follows:
        - GA causes instability and leads to failure of unlearning.
        - But GD is stable throughout training, which results in a memorized model $\theta_{mem}$.
        - Given the intuition that forgetting is the opposite of memorization, we can use TV to reach forgetting based on $\theta_{mem}$. (`Failure of GA`$\rightarrow$`Use GD to memorize`$\rightarrow$`memorize to forget via TV`.)
        - Therefore, our philosophy focused on addressing GA, and TV is just a realization of our idea. We believe other realizations, such as Gradient Surgery, Projected Gradient Adversary, and Influence-Function-based Reversal, could also be helpful. But our point is that MOX is philosophically different from TV.

**Minor Concerns**: We have also carefully addressed some minor concerns.
- We justified the unlearning performance of MOX under different modalities.
- We validated the parameter sensitivity and provided clear guidelines to choose $\alpha$.
- We provided theoretical justification for the stability of GD, and intuitively explained the condition of MOX (a well-pretrained model possesses NTK/Linear property in local space).
- We generalized MOX to real-time forgetting and validated its independence of the retain dataset.
- Moreover, though many reviewers recognized our novelty and significance, we stress that GA is still one main strategy for unlearning, as it has been frequently studied in very recent works.


We deeply appreciate everyone who has been involved in the review process. This paper could not have been further improved without your efforts. We are confident that all concerns have been reasonably and convincingly addressed. Should there be any questions from the AC or SAC, we are also more than happy to provide further assistance.

Hope this summary could be helpful for the new AC or SAC. Thank you for your contribution to the ICLR community.

Best wishes,
Authors.

---

### Meta-Review · Area_Chair_T4Pq · 2026-01-17

**Summary:**

The paper proposes a method for LLM unlearning with a two-stage process: It first reinforces memorization on forget data; then applies a linear extrapolation from this memorization model and the original reference model to derive the unlearned model.

Reviewers expressed the following concerns:
- The paper provides limited theoretical explanation of how and when the proposed memorization model achieves unlearning.
- Experiments suggest the method is sensitive to hyperparameters
- The paper has limited novelty as its core design is close to the Task Vector method.
- The paper lacks comparisons of computational efficiency and resource costs.

**Reviewer Concerns:**

Authors provided detailed responses to reviewer concerns.
A clear theoretical explanation/justifiication of why and when the proposed method achieves forgetting is still missing.
The discussion on novelty over task vector-based methods requires careful review.

Overall, this is a nice paper but requires some major changes.

**Reviewer Scores:**

Reviewer scores are 6,8,4,4

One reviewer engaged with the authors in the rebuttal period, but seemed unconvinced by authors response.
It is difficult to tell if the reviewers would have changed their scores.

---

### Decision · Program_Chairs · 2026-01-26

Reject